



**Ecosystem-specific patterns and drivers of global reactive iron**
**mineral-associated organic carbon**
Bo Zhao[1], Amin Dou[1], Zhiwei Zhang[1], Zhenyu Chen[1], Wenbo Sun[1], Yanli Feng[1],
Xiaojuan Wang[2], Qiang Wang[1,*]
*[1]State Key Laboratory of Herbage Improvement and Grassland Agro-ecosystems,*
*College of Pastoral Agriculture Science and Technology, Lanzhou University, Lanzhou*
*730000, China*
*[2]Natural History Research Center, Shanghai Natural History Museum, Shanghai*
*Science & Technology Museum, 200127 Shanghai, China*
*[*]**Corresponding author:** Qiang Wang (Phone: +86-136-6933-7869; Email:
wqiang@lzu.edu.cn)
**Type of paper:** Research Paper; Text pages: 24; Figures: 7





## Abstract

Reactive iron (Fe) oxides are vital for long-term soil/sediment organic carbon (SOC) storage. However, the patterns and drivers of Fe-associated organic carbon (Fe-OC) over global geographic scales under various ecosystem types remain largely controversial. Here, we provided for the first time a systematic assessment of the distribution patterns and determinants of Fe-OC content and its contribution ($f$Fe-OC) by assembling a global dataset comprising 862 observations from 325 sites in distinct ecosystems. We found that Fe-OC content across global ecosystems ranged from 0 to 83.3 g/kg ($f$Fe-OC ranged from 0 to 82.4%), reflecting the high variability of the Fe-OC pool. Fe-OC contents varied with ecosystem type, being greater in wetlands with a high molar ratio of Fe-OC/dithionite-extractable Fe ($Fe_d$) compared with marine and continental ecosystems. Furthermore, $f$Fe-OC in wetlands was significantly lower than that in other ecosystems due to rich OC. In contrast with climate variables and soil pH, the random forest modelling and multivariate analysis showed that the Fe-OC:$Fe_d$ and SOC were the predominant predictors of Fe-OC content and $f$Fe-OC in wetlands and continents, whereas $Fe_d$ content was a primary driver in marine ecosystems. Based on upper estimates of global SOC storage in various ecosystem types, we further estimated that $83.84 \pm 3.86$ Pg, $172.45 \pm 8.74$ Pg, and $24.48 \pm 0.87$ Pg of SOC were preserved by association with Fe oxides in wetlands, continental and marine ecosystems, respectively. Taken together, our findings highlighted the importance of reactive Fe oxides in global SOC preservation, and their controlling factors were ecosystem-specific.

**Keywords:** Ecosystem type, mineral protection, reactive iron oxides, iron-bound





organic carbon, organic carbon preservation





## 1. Introduction

The global soil (sediment) organic carbon (SOC) cycle has become one of the hotspots in biogeochemical and global climate change research (Lal, 2004a; Crowther et al., 2016). Organic carbon (OC) sequestration is a significant ecosystem service (such as climate mitigation, soil fertility and ecosystem stability, etc.) provided by terrestrial, wetland, and marine ecosystems. Accumulating evidence has shown that the reactive mineral matrix plays a critical role in sequestering and stabilizing SOC (Kramer and Chadwick, 2018; Ye et al., 2022). OC has a strong affinity for reactive Fe (hydr-)oxides (Longman et al., 2022), and the resulting Fe and OC association by adsorption or coprecipitation is thought to promote OC long-term preservation in soils and sediments (Schmidt et al., 2011; Hemingway et al., 2019). Therefore, a systematic understanding of the patterns and drivers of Fe-associated OC (Fe-OC) is pivotal for accurately predicting SOC dynamics and reducing model uncertainties in forecasting carbon-climate feedback at the global scale.

In comparison to other metal minerals, Fe (hydr-)oxides, one of the most prevalent reactive minerals, have larger specific surface areas, a higher OC affinity, and a greater potential to retain SOC (Guggenberger and Kaiser, 2003; Eusterhues et al., 2005; Kaiser et al., 2007). A growing body of studies has suggested that Fe (hydr-)oxides play a fundamental role in stabilizing SOC in sediment and soil (Yu et al., 2021). Recently, Fe-OC has been extracted and quantified through the bicarbonate-citrate-dithionite (BCD) method, and was estimated to constitute 21.5% (Lalonde et al., 2012), 4.7–37.8% (Zhao et al., 2016; Fang et al., 2019; Zong et al., 2021), and 3.4–11.8% (Huang et al.,



2021; Wang et al., 2021) of SOC in marine sediments, continents (i.e., forests,
grasslands, farmland) and wetlands (i.e., coastal, peatland, and lake wetlands),
respectively. The Fe-OC content and contribution ($f$Fe-OC) vary with ecosystem type.
Marine sediments are the largest OC sink on Earth and are crucial to the global carbon
cycle. Reactive Fe minerals can protect and bury large amounts of SOC within marine
sediments, constituting a "rusty sink" (Lalonde et al., 2012). The $f$Fe-OC in marine
sediments is significantly lower than that in offshore estuarine sediments due to
differences in sediment mineralogy, reactive Fe source and organic matter composition
(Longman et al., 2022). It is well known that wetland ecosystems possess an extremely
high rate of OC sequestration (McLeod et al., 2011; Hopkinson et al., 2012). Compared
with continental and marine ecosystems, wetland soils or sediments are periodically
submerged due to (semi)diurnal tidal cycles or fluctuations in the water table (Yu et al.,
2021). Thus, in wetland environments, Fe (hydr-)oxides are repeatedly formed and
destroyed as a result of periodical redox-induced changes in $Fe^{2+}/Fe^{3+}$ (Patzner et al.,
2020), which is thought to weaken the interaction between Fe and OC (Huang and Hall,
2017; LaCroix et al., 2019; Anthony and Silver, 2020). However, Wang et al. (2017)
proposed an important "iron gate" mechanism in OC-rich wetlands (Wang et al., 2017),
and showed that the contribution of Fe-OC to SOC ($f$Fe-OC) in wetlands and uplands
is equally important (Wang et al., 2017). Thus, a systematic analysis of Fe-OC content
and $f$Fe-OC in continental, wetland and marine ecosystems at the global scale can
provide evidence for the importance of reactive Fe minerals in global climate change.

Recently, some studies have found that the Fe-OC content and $f$Fe-OC are mainly



controlled by soil properties (Grybos et al., 2009; Ye et al., 2022), organic matter
composition (Fisher et al., 2020), and climate (or latitude) (Kramer and Chadwick,
2018). For instance, Fe-OC increases with increasing latitude, mean annual
precipitation (MAP), SOC content, and potential evapotranspiration (Zhao et al., 2016;
Kramer and Chadwick, 2018), but it decreases with increasing soil pH at the continental
scale (Ye et al., 2022). However, Fe-OC content and $f$Fe-OC in farmland soils are not
related to latitude, mean annual temperature (MAT) and MAP but are related to SOC
content (Wan et al., 2019). In peatlands, Huang et al. (2021) found that Fe-OC content
is positively correlated with the SOC content, C:N, and MAT but not with MAP at the
regional scale (Huang et al., 2021). However, Fe-OC in coastal wetlands was positively
correlated with amorphous Fe content and clay content, but negatively correlated with
soil pH and phenol oxidase activity (Bai et al., 2021). In marine sediments, Fe-OC
content may be mainly responsible for SOC content and organic matter functional
groups (especially carboxyl content) (Wang et al., 2019; Fisher et al., 2020).
Additionally, according to Kramer & Chadwick (2018), $f$Fe-OC in humid climate forest
regions was much higher than that in semiarid and arid regions, confirming the natural
linkages between $f$Fe-OC and climate (Kramer and Chadwick, 2018). Fe-OC content
is also influenced by the bonding mechanism of Fe and OC (Wagai and Mayer, 2007;
Faust et al., 2021). The bonding mechanism between Fe and OC is determined by the
Fe-OC/dithionite-extractable Fe (Fe$_d$) molar ratio (Faust et al., 2021; Wang et al., 2021),
with less than 1 indicating an Fe-OC bonding form of monolayer surface sorption, and
greater than 6 indicating a bonding mechanism dominated by coprecipitation (Wagai



and Mayer, 2007; Lalonde et al., 2012). Generally, the OC content of the complexes

obtained by coprecipitation is much higher than that of adsorption (Chen et al., 2014),

which may also explain the wide variations in Fe-OC and $f$Fe-OC. Thus, uncovering

the factors controlling Fe-OC formation/association at the global scale is a prerequisite

for predicting the size of the OC pool and its feedback on global climate change.

However, the determinants of Fe-OC associations remain unknown globally, and only

two studies on Fe-OC have been undertaken at continental scale, which focus on the

relationships of Fe-OC and soil pH (Ye et al., 2022), MAP and potential

evapotranspiration (Kramer and Chadwick, 2018). These studies overlooked the

influence of climate and soil properties (such as soil pH, $Fe_d$, Fe-OC:$Fe_d$, clay content)

in controlling Fe-OC and $f$Fe-OC in wetland and marine ecosystems. Furthermore, they

have not yet explored the relationship between these key factors and Fe-OC and $f$Fe-

OC across global ecosystem types. A deeper understanding of these limitations in

continental, wetland and marine ecosystems will allow us to draw clear conclusions

regarding global patterns and drivers of Fe-OC.

In this study, we provide a comprehensive analysis of the spatial variability and

characteristics of Fe-OC among continental, wetland and marine ecosystems and its

governing factors globally. Specifically, we analysed data from 862 observations from

46 published papers and the National Ecological Observatory Network (NEON) to

explore (i) the importance of Fe-OC to SOC storage in wetland and marine ecosystems

and its level compared with continental ecosystems, (ii) whether the distribution

patterns (i.e., spatial variability) of Fe-OC and the relationships between key factors


and Fe-OC differ among ecosystem types? (iii) The bonding mechanism of reactive Fe
and OC in different ecosystem types, i.e., adsorption or coprecipitation?
**2. Materials and methods**
**2.1 Study selection**
The ecosystem types included continents, wetlands, and marine ecosystems in this
study. We conducted extensive literature searches on the Web of Science
(https://www.webofscience.com) and China National Knowledge Resource Integrated
databases, and searched for relevant research published from 2010 to August 2022. The
appropriate studies were identified by the following search terms: ('reactive mineral'
OR 'iron') AND ('bound' OR 'associated' OR 'stabilization' OR 'interaction' OR
'sequestration') AND ('organic carbon')) (Fig. S1). The following criteria must be met
for inclusion in this study: (a) soil samples at 0-100cm depth must be collected from in
situ observation data of wetlands (i.e., peatland, bog, fen, deltaic, lake wetland,
mangrove wetland, and estuary wetland), forests (i.e., evergreen forest, and deciduous
forest), grasslands (i.e., temperate grasslands and alpine grasslands), farmland (i.e.,
paddy field and crop), and marine ecosystems (i.e., marine and river sediments); (b) the
contents of Fe-OC and $Fe_d$ were measured using the BCD method in bulk soil; and (c)
Fe-OC, Fe-OC/$Fe_d$ molar ratio must be provided or could be calculated from the
publications. In total, we compiled 862 data records from 46 published papers, along
with 42 additional data collected from NEON. The dataset involved 325 sites, with
latitudes between 25.22°S and 81.75°N and longitudes between 156.4°W and 174.4°E
(Fig. 1).



## 2.2 Data assembly and collection

Data from published articles and NEON were assembled to construct the Fe-OC dataset. Site-specific data such as ecosystem type, MAP, MAT, latitude, longitude, clay, soil pH, SOC, Fe-OC, $Fe_d$, Fe-OC/$Fe_d$ molar ratio, and $f$Fe-OC (calculated using the following equation: $f$Fe-OC (%) = Fe-OC/SOC×100%) were collected from each published paper; other details are shown in Table S1. If the MAT and MAP are not reported, the data for each site shall be obtained from the WordClim database (http://www.worldclim.org.d). All original data and average data were taken from the published articles' text, graphs, and tables. When data were presented graphically, the numerical data were digitized and extracted with the GetData Graph Digitizer (version 4.4).

## 2.3 Statistical analysis

All data analyses were conducted using the R platform (v 4.1.2; https://www.r-project.org/). We used the Shapiro-Wilk test to determine the homogeneity of variances and the normal distribution of the data before using parametric methods. We used the Kruskal–Wallis test to determine significant differences among different ecosystems.

Hedges' g, a bias-corrected standardized mean difference, was used to measure effect size to account for the bias of ecosystem-scale Fe-OC associated with small sample sizes (Chien & Krumins, 2022; Smale et al., 2020). Based on ecosystem types, all data were divided into continental, marine and wetland ecosystems, and the data were averaged separately for each ecosystem, representing 'control'. The sample sizes of individual cases (i.e., a single published article) represent 'treatment'. The




standardized mean difference between the 'control' and 'treatment' was measured by
the pooled variance (Chien & Krumins, 2022). We used the package "metafor" in R (v
4.1.2; https://www.r-project.org/) to generate forest plots for every ecosystem by using
a random effects model (Fig. S2). We calculated the total observed change ($I^2$) and used
heterogeneity test (Q) to verify the heterogeneity of the collected data, and an $I^2$ value
higher than 75% or $p < 0.05$ indicates substantial heterogeneity (Meisner et al., 2014).
We performed Spearman's correlation analyses to evaluate the relationship between
environmental variables (SOC, MAT, MAP, clay, soil pH, Fe-OC:Fe$_d$, Fe$_d$, and latitude)
and Fe-OC and $f$Fe-OC. The linear ("lm" function in R) fitting was demonstrated to
analyse the relationships between environmental variables and Fe-OC and $f$Fe-OC. The
significant correlation was considered at $p < 0.05$. To test the relative importance of
these drivers, a random forest analysis (RF, Breiman, 2001) was performed according
to the protocol described by Delgado-Baquerizo et al. (2016). For the RF analyses, the
climate variables (MAT, MAP), soil properties (SOC, clay, soil pH, Fe-OC:Fe$_d$, and
Fe$_d$), and geographical location (i.e., latitude) were involved as predictors, and the Fe-
OC and $f$Fe-OC changes and dynamics as the response variables. The significance of
the models and cross-validated $R^2$ values were evaluated with 500 permutations of the
response variables with the "A3" R package. Similarly, using the "rfPermute" package
for R ($p < 0.05$), the importance of each predictor on the response variables was
evaluated.





## 3. Results

### 3.1 Fe-associated OC and its related indicators across ecosystem types

Across global ecosystem types (i.e., continental, wetland and marine ecosystems), Fe-OC content (n = 862) and $f$Fe-OC (n = 855) varied significantly and ranged from 0 to 83.3 mg g$^{-1}$ (mean: 5.62 ± 0.32 mg g$^{-1}$) and 0–82.4% (mean: 16.03± 0.41%), respectively (Figs. 2a, b). The contents of Fe-OC in continental, marine and wetland ecosystems were 5.42 ± 0.41 mg g$^{-1}$ ($f$Fe-OC: 17.76 ± 0.90%), 2.34 ± 0.12 mg g$^{-1}$ ($f$Fe-OC: 16.32 ± 0.58%) and 9.97 ± 0.91 mg g$^{-1}$ ($f$Fe-OC: 13.70 ± 0.63%), respectively, with significant differences among ecosystem types ($p < 0.05$; Fig. 2a), but no significant difference in $f$Fe-OC was observed ($p > 0.05$; Fig. 2b). Meanwhile, Hedges' g unbiased standardized mean difference showed that small sample sizes at local scale (i.e., single published articles) had obvious distinct effect sizes for ecosystem-scale Fe-OC ($I^2 > 75\%$ or $p < 0.05$), especially for marine ecosystems (Fig. S2). Fe$_d$ contents (n = 856) ranged from 0.03 to 245 mg g$^{-1}$ (mean: 9.43 ± 0.53 mg g$^{-1}$; Fig. 2c); that is, Fe$_d$ varied 8167-fold, which was significantly higher in continental ecosystems than in wetland and marine ecosystems ($p < 0.05$; Fig. 2c). The Fe-OC/Fe$_d$ molar ratio (n = 855) ranged from 0–331.68 (mean: 8.40 ± 0.85) at the global scale, and its mean value was significantly higher in wetlands than in continents, while the minimum value was found in marine systems ($p < 0.05$; Fig. 2d). SOC contents (n = 854) ranged from 0.3 to 423.74 mg g$^{-1}$ (mean: 43.28 ± 2.52 mg g$^{-1}$), which had similar changes with the Fe-OC contents among ecosystem types ($p < 0.05$; Fig. 2e). Taken together, the Fe-OC, SOC, Fe-OC/Fe$_d$ molar ratio, and soil pH were significantly higher



in wetlands, with the lowest values in marine ecosystems across global ecosystem types.

**3.2. Effect of environmental factors on Fe-OC and $f$Fe-OC across ecosystem types**

We analysed their relationships with climate variables and soil properties to better
understand the potential effect factors behind the observed variance in Fe-OC contents
and $f$Fe-OC among ecosystem types (Fig. 3). Among them, in wetland ecosystems, Fe-
OC content showed a negative correlation with MAT ($R = -0.42$, $p < 0.001$; Fig. 3a) and
MAP ($R = -0.26$, $p < 0.001$; Fig. 3b), while $f$Fe-OC was positively correlated with the
climate variables (MAT, MAP) (Figs. 3i, j). The Fe-OC content decreased significantly
with increasing soil pH in wetlands ($R = -0.24$, $p < 0.01$; Fig. 3c) and continents ($R = -$
$0.19$, $p < 0.05$; Fig. 3c), but $f$Fe-OC increased with increasing soil pH ($R = 0.52$, $p <$
$0.001$; Fig. 3k) in wetlands. Across the ecosystem types, $Fe_d$ contents showed positive
correlations with Fe-OC ($R = 0.25$, $p < 0.001$; Fig. 3g) and $f$Fe-OC ($R = 0.28$, $p < 0.001$;
Fig. 3O) in marine ecosystems only. Moreover, Fe-OC increases significantly with $Fe_d$
contents ($R = 0.35$, $p < 0.001$; Fig. 3g) in wetlands, but $f$Fe-OC does not; however, $Fe_d$
content has no relationship with Fe-OC and $f$Fe-OC in continental ecosystems. The
molar ratio of Fe-OC/$Fe_d$ was positively correlated with Fe-OC and $f$Fe-OC in three
ecosystem types, except for $f$Fe-OC in wetlands (Figs. 3e, m). Fe-OC contents
increased significantly, but $f$Fe-OC (except marine) decreased with increasing SOC
contents in all ecosystems (Figs. 3f, n). At continental scales, Fe-OC content ($R = 0.35$,
$p < 0.001$; Fig. 3d) and $f$Fe-OC ($R = 0.44$, $p < 0.001$; Fig. 3l) were positively related to
clay content. Latitudinal patterns in Fe-OC content and $f$Fe-OC were observed across
global ecosystem types (Figs. 3h, p). Taken together, Fe-OC contents are significantly





correlated with both SOC and the Fe-OC/Fe$_d$ molar ratio, which may be important
predictors of Fe-OC in global ecosystems.

Moreover, according to RF analysis, the Fe-OC/Fe$_d$ molar ratio and SOC and Fe$_d$

contents were found to be the most important variables for predicting the Fe-OC content
and $f$Fe-OC across global ecosystem types (Fig. 4). Different controlling factors on Fe-
OC content and $f$Fe-OC were operational among ecosystem types. At continental scales,
the Fe-OC/Fe$_d$ molar ratio was a central driver of the Fe-OC content and $f$Fe-OC, and
the contents of SOC and Fe$_d$ had a more significant influence than the soil pH and
climate variables (MAT, MAP) (Figs. 4a, b). The largest influence on Fe-OC content
and $f$Fe-OC in marine ecosystem was in the order of Fe$_d$ > Fe-OC:F$_d$ > SOC > latitude
(Figs. 4c, d). In wetlands, the Fe-OC/Fe$_d$ molar ratio was the main driver of Fe-OC,
whereas SOC had a more significant role than Fe$_d$ and soil pH (Fig. 5e); For $f$Fe-OC,
the largest influence was in the range of SOC > Fe-OC:F$_d$ > pH > MAT > MAP > Fe$_d$
(Fig. 4f). The role of Fe$_d$ content in controlling Fe-OC content and $f$Fe-OC was greater
in marine systems than in continents and wetlands. These results revealed that drivers
of both Fe-OC content and $f$Fe-OC were ecosystem specific. The climate predictors
accounted for relatively small percentages in all ecosystems. Collectively, Fe-OC:F$_d$,
SOC, and Fe$_d$ were all selected by RF analysis as important predictors of changes in
Fe-OC content and $f$Fe-OC, which agreed with the results of our Spearman's
correlation analyses (Fig. 3).
**3.3. The vital role of Fe-OC:Fe$_d$ in controlling Fe/OC interactions**

At the continental scale, the proportions of Fe-OC/Fe$_d$ molar ratios less than 1 (<



1), between 1 and 6 (1–6), and higher than 6 (> 6) were 33.10%, 47.89%, and 19.01%,
respectively (Fig. 5). Moreover, we found that the proportions of 1–6 were larger in
grasslands and farmland than in forests, but the proportions of > 6 in grasslands were
higher. In marine ecosystems, the proportion of Fe-OC:$Fe_d$ < 1 (31.0%) is lower than
that of 1–6 (63.75%), and the proportion of > 6 (5.31%) is the smallest. However, the
proportion of Fe-OC:$Fe_d$ > 6 (39.44%) in wetlands was significantly higher than that in
other ecosystems (19.01% and 5.31%, respectively), but the proportion of <1 (13.55%)
was lower.

Consistent with our expectation, the molar ratio was significantly positively

correlated with Fe-OC and SOC contents but negatively correlated with $Fe_d$ in all
ecosystem types (Fig. 6). Moreover, the results showed that MAT and MAP are also
major negative regulators of the molar ratio dynamics at the continental scale, whereas
in wetlands, it is soil pH (Figs. 6a, c).
**4. Discussion**
**4.1 Reactive Fe promotes SOC preservation at the global scale**

In contrast to previous studies (Kramer and Chadwick, 2018; Yu et al., 2021; Ye et

al., 2022), our findings suggested that a comprehensive analysis of global patterns of
Fe-OC associations across ecosystem types, particularly in wetland and marine
ecosystems, can bridge the knowledge gap in understanding the importance of global
SOC preservation by reactive Fe. Generally, mineral-associated organic carbon is the
dominant SOC pool in soil systems, with a proportion of approximately 50–80% of
SOC (Cotrufo et al., 2019). As an important component of reactive minerals, Fe



(hydr-)oxides play a fundamental role in the formation and dynamics of mineral-
associated organic carbon (Lalonde et al., 2012). Our findings showed that the average
content of Fe-OC was $5.63 \pm 0.32$ mg g$^{-1}$ soil (n = 862), and the proportion ($f$Fe-OC)
of Fe-OC in total SOC was $16.03 \pm 0.41\%$ (n = 855) across global ecosystems (Figs.
2a, b), indicating that Fe-OC is essential to the persistence of SOC. Consistent with
our expectation, significant difference in $f$Fe-OC was observed among different
ecosystem types. At the continental scale, the mean $f$Fe-OC was $17.75 \pm 0.90\%$ (0–
82.36%, n = 284), which was consistent with findings from Tibetan alpine meadows
($15.8 \pm 12.0\%$) (Fang et al., 2019) but was lower than those for continental-scale
forests, such as moist forests (25.3–49.8%) and wet forests (47.1–64.1%) (Zhao et al.,
2016; Kramer and Chadwick, 2018). According to upper estimates of global continent
SOC storage (971 Pg) (including forest (383 Pg), grassland (423Pg) and farmland (165
Pg)) (Carter et al., 2000; Lal, 2004b; Pan et al., 2011; Prentice et al., 2001), we
estimated that $172.45 \pm 8.74$ Pg of SOC was bound to Fe oxides in continental
ecosystems. Meanwhile, we predicted that $49.02 \pm 5.24$ Pg ($12.80 \pm 1.37\%$), $74.28 \pm$
4.95 Pg ($17.56 \pm 1.17\%$), and $28.41 \pm 4.34$ Pg ($17.22 \pm 2.63\%$) of SOC were associated
with Fe oxides in forests, grasslands, and farmlands, respectively. In contrast to
continental ecosystems, evidence of interactions between Fe and SOC in marine
sediments has been reported more often (Berner, 1970), but the potentially importance
of reactive Fe for SOC preservation has only recently been recognized in marine
sediments (Lalonde et al., 2012). Recently, an accumulating body of studies have
shown that reactive Fe has a strong affinity for SOC, forming stable Fe-OC complexes



that can persist for thousands of years in marine sediments, serving as a "rusty sink"
for marine sedimentary carbon (Lalonde et al., 2012; Faust et al., 2021). Our findings
suggested that $f$Fe-OC in global marine sediments ranged widely from 0.51% to
60.3%, with a mean of 16.32 ± 0.58%. These values are consistent with published
estimates for the East China Sea (13.2 ± 8.9%) (Ma et al., 2018), Bohai Sea (11.5 ±
8.3%) (Wang et al., 2019), River Delta (8.1–20.2%) (Shields et al., 2016), Barents Sea
(10–20%) (Faust et al., 2021), and global marine surface sediments (21.8 ± 8.6%)
(Lalonde et al., 2012). Based on model-predicted global marine sedimentary OC
stocks (150 Pg) (Hedges & Keil, 1995), we further estimated that 24.48 ± 0.87 Pg of
the marine sedimentary OC was directly associated with Fe oxides, which was
comparable to the results of previous study (19–45 Pg OC) (Lalonde et al., 2012).
Wetland ecosystems, however, frequently experience seawater flooding, atmosphere
exposure, and/or disruption of the hydrological balance due to (semi)diurnal tidal
cycles or water table drawdown, in contrast to continental and marine systems (Huang
and Hall, 2017; Patzner et al., 2020). Fe-OC associations are weakened with the
reductive breakdown of Fe(III) (hydr)oxides driven by periodic soil redox processes
(Patzner et al., 2020). Although wetlands store 20–30% of the Earth's soil carbon
(~2500 Pg) (Roulet, 2000; Bridgham et al., 2006), the importance of Fe-OC in wetland
soils/sediments remains controversial. In global wetlands, we found that the absolute
content of Fe-OC was significantly higher than those in continental and marine
ecosystems, whereas the opposite was true for $f$Fe-OC, which was significantly lower
in wetlands. Our findings in wetlands were also consistent with those of Ye et al. (2022)



at continental scales (13.6 ± 1.0%; Ye et al., 2022) and regional-scale wetlands (16.1
± 1.4%) (Wang et al., 2021) but were higher than those for specific peatland
ecosystems (3.42 ± 1.32%) (Huang et al., 2021). Compared with coastal wetlands (for
instance, mangrove wetland and tidal wetland) (Bai et al., 2021; Zhao et al., 2022),
inland wetlands (for instance, alpine wetland and peatland) have lower $f$Fe-OC (Wang
et al., 2017; Huang et al., 2021), which may lead to significantly lower $f$Fe-OC in
global wetlands. Therefore, the significance of reactive Fe minerals for SOC
sequestration in global wetlands may be underestimated based on peatland $f$Fe-OC
(Huang et al., 2022). Here, based on global wetland $f$Fe-OC and total SOC stocks (612
Pg) (Yu et al., 2010), we predicted that 83.84 ± 3.86 Pg of SOC was preserved by
binding to Fe oxides. Collectively, these findings confirmed the fundamental role of
reactive Fe minerals for OC sequestration and conservation in global ecosystems.

Two possible mechanisms may explain the higher Fe-OC content in wetlands than

in other ecosystems. First, the molar ratios of Fe-OC:$Fe_d$ were significantly higher in
wetlands than in continental and marine ecosystems ($p < 0.05$; Fig. 2d), suggesting that
in wetlands reactive Fe is more effective in OC binding (Wagai and Mayer, 2007; Riedel
et al., 2013). Numerous studies have shown that the Fe-OC:$Fe_d$ acts as an indicator of
Fe/OC interaction types (Lalonde et al., 2012; Wang et al., 2017), with <1 suggesting
that the OC-Fe bonding form is dominated by simple mono-layer adsorption, while
higher ratios indicating coprecipitation (Wagai and Mayer, 2007; Faust et al., 2021).
Thus, compared with other ecosystems, in wetlands coprecipitation played a more
significant role in the binding/association of Fe-OC. Second, the SOC content in





wetlands was significantly higher than that in continental and marine ecosystems ($p <$
0.05; Fig. 2e), and it is generally believed that the SOC in wetlands has various
chemical bonds or chemical compositions (Wang et al., 2017; Coward et al., 2018).
Thus, the high SOC content in wetlands could be responsible for the predominance of
Fe(II) with a strong OC-complexation capacity (Jones et al., 2015; Bhattacharyya et al.,
2018), especially the enrichment of phenolic (Freeman et al., 2001), ultimately
promoting the Fe-OC association (Riedel et al., 2013; Coward et al., 2018).

**4.2 Ecosystem-specific relationships of Fe-OC associations with key factors**

The role of soil pH, SOC, $Fe_d$, Fe-OC:$Fe_d$, MAT and MAP in controlling Fe-OC
contents and $f$Fe-OC among ecosystem types was thoroughly analysed. A compilation
of global datasets including continental, wetland, and marine ecosystems demonstrated
that Fe-OC content and $f$Fe-OC are strongly coupled to both the Fe-OC:$Fe_d$ molar ratio
and SOC content ($p < 0.001$; Figs. 3e, f, m, n), indicating that the two variables are
important determinants of Fe-OC content and $f$Fe-OC. The results from the RF models
also revealed that Fe-OC:$Fe_d$ molar ratio, SOC content, and $Fe_d$ content were important
predictors of Fe-OC and $f$Fe-OC across ecosystem types (Fig. 4). Collectively, these
findings suggested a generic dependency of Fe-OC and $f$Fe-OC on the Fe-OC:$Fe_d$
molar ratio and SOC, regardless of their ecosystem types. Former studies on the
response of Fe-OC to climate variables and soil properties only concentrated on the
continental scale and specific ecosystems with limited data (Ye et al., 2022), making it
challenging to reach definitive conclusions. Kramer & Chadwick (2018) concluded that
continental-scale Fe-OC variation depended on MAP and potential evapotranspiration





but overlooked the role of soil properties (Kramer and Chadwick, 2018). Our findings
further showed that the soils with higher MAP were linked with lower soil pH (Fig. 6),
which had a positive effect on Fe-OC contents at the continental scale (Fig. 3c), and
these results are in line with Ye et al. (2022) (Ye et al., 2022). Furthermore, we found
that Fe-OC content was primarily controlled by the Fe-OC:$Fe_d$ molar ratio at the
continental scale and wetlands (Fig. 7). Given the strong affinity of OC with [Fe(III)]
(hydr-)oxides, we speculated that an increase in $Fe_d$ content would lead to higher Fe-
OC content, assuming sufficient SOC was present (Ma et al., 2018; Wang et al., 2019).
Although reactive Fe plays a fundamental role in OC binding, its content is not related
to Fe-OC content in specific terrestrial ecosystems, such as the Qinghai-Tibet Plateau
and regional-scale forests (Mu et al., 2016; Zhao et al., 2016). Our study, for the first
time, illustrated the crucial role of $Fe_d$ in controlling Fe-OC contents and $f$Fe-OC in
global marine ecosystems (Fig. 3g and Fig. 4c). Previous findings indicated that
increased terrigenous reactive Fe inputs contributed to higher Fe-OC contents (Ma et
al., 2018; Wang et al., 2019). Therefore, sedimentary $Fe_d$ content was the controlling
factor of Fe-OC associations in marine ecosystems. The findings of Faust et al. (2021),
however, who showed that a higher $Fe_d$ content does not always enhance Fe-OC
associations in Arctic marine sediments, were in contrast to our findings (Faust et al.,
2021). The differences between our results and those of Arctic marine sediments may
be mainly related to the study scale. Nevertheless, the bonding mechanism of Fe and
OC (adsorption vs. coprecipitation) is a predominant driver of $f$Fe-OC in wetlands and
continental ecosystems, as illustrated by the RF analysis and a good linear correlation.





Given that the Fe and OC interactions are substantially controlled by Fe redox processes (Riedel et al., 2013; Adhikari et al., 2016), we posited that the contents and proportions of Fe-OC are governed mainly by Fe redox cycling and associated bonding mechanisms, with the exception of the marine ecosystems. The results of this study suggested that future climate warming may increase the proportions of Fe-OC in the total SOC, especially in wetlands (Figs. 3i, j), even though additional research is necessary to fully understand the effects of climate changes on Fe-OC at the global scale.

**4.3 Potential bonding mechanism between Fe and OC across ecosystem types**

Adsorption and coprecipitation are well-known to be important and well-documented processes for the association of OC and reactive Fe (Lalonde et al., 2012; Chen et al., 2014). Reactive Fe can act as sorbents of OC to adsorb large amounts of OC to mineral surfaces due to its ubiquity in the environment, high surface area and small particle size (Kaiser and Guggenberger, 2003). Riedel et al. (2013) showed that coprecipitated Fe-OC complexes form when reduced Fe is oxidized in the presence of dissolved OC at the oxic-anoxic interface and present a high Fe-OC:$Fe_d$ molar ratio (Riedel et al., 2013). The Fe-OC:$Fe_d$ molar ratio can be used as an indicator for the bonding mechanism between Fe and OC (Lalonde et al., 2012; Peter and Sobek, 2018; Faust et al., 2021; Wang et al., 2021), with <1 indicating simple mono-layer sorption and >6 indicating coprecipitation (Tipping et al., 2002; Wagai and Mayer, 2007). Our findings suggested that the average Fe-OC:$Fe_d$ molar ratio was $10.50 \pm 1.91$ at the continental scale. However, we could see that the Fe-OC:$Fe_d$ molar ratios (mean 70.18 $\pm$ 13.82; range 2.58–331.68) were much higher in permafrost regions of the Tibetan





Plateau than in other specific terrestrial ecosystems, resulting from relatively high Fe-
OC and low $Fe_d$ (Mu et al., 2016). In view of the very high molar ratio, coprecipitation
is the dominant bonding mechanism of OC and Fe, which contributes to $f$Fe-OC
reaching 59.5% (average $19.5 \pm 12.3\%$) in Fe-poor (range $0.03-2.68$ mg $g^{-1}$ soil)
permafrost soils of the Tibetan Plateau (Mu et al., 2016). If the permafrost region of the
Tibetan Plateau is excluded, the Fe-OC:$Fe_d$ molar ratio in global terrestrial ecosystems
was only $3.74 \pm 0.47$, indicating that coprecipitation will become a less important
bonding mechanism. Recently, a regional-scale survey including typical grasslands,
shrublands and forests by Wang et al. (2021) reported that the average Fe-OC:$Fe_d$ molar
ratio was $3.0 \pm 0.5$ (Wang et al., 2021), which lends further credence to the findings
mentioned above. The average Fe-OC:$Fe_d$ molar ratio was $2.56 \pm 0.19$ (n = 320; range
$0.04–31.59$) in global marine ecosystems, similar to that of the Bohai Sea ($1.59 \pm 1.37$)
(Wang et al., 2019), the Southern Yellow Sea ($1.68 \pm 1.80$) (Ma et al., 2018), East China
Sea ($1.53 \pm 1.28$) (Ma et al., 2018), and Barents Sea ($2.56 \pm 1.76$) (Faust et al., 2021),
but was much lower than the previous average of global oceans (n = 42; $6.10 \pm 7.5$)
(Lalonde et al., 2012), Arctic shelf (Salvadó et al., 2015), and intermediate/old river
delta (Shields et al., 2016). Moreover, in wetlands, the molar ratios of Fe-OC:$Fe_d$ were
higher ($13.47 \pm 1.81$) than those in continental and marine ecosystems. These results
were in accordance with previous findings in regional-scale wetlands ($12.78 \pm 2.43$)
(Wang et al., 2021) and coastal wetlands ($11.0 \pm 4.5$) (Bai et al., 2021) but higher than
that peatlands (mean 6.53) (Huang et al., 2021). This suggested that the interaction
between OC and Fe in wetland ecosystems is mainly dominated by coprecipitation at



the global scale, with a molar ratio of >6 usually. Overall, across the global ecosystem
types, the average proportion of Fe-OC:$Fe_d$ > 1 ranged from 60 to 80% (Fig. 5), which
indicated the importance of both adsorption and coprecipitation interactions.
Furthermore, we found that SOC content could enhance the molar ratio of Fe-OC:$Fe_d$
by positively regulating Fe-OC content. At the continental scale, climate variables
(MAT, MAP) can negatively regulate the molar ratio by changing the $Fe_d$ content (Fig.
6a), while in wetlands, soil pH changes the Fe-OC content and then negatively regulates
the molar ratio (Fig. 6c). Despite the molar ratio being widely used as an important
indicator of the bonding mechanism of Fe and OC, recent studies have shown that only
a portion of reactive Fe (25.7–62.6%) was directly associated with OC (Barber et al.,
2017). Thus, using the raw Fe-OC:Fe molar ratio may result in an underestimation of
the actual molar ratio due to the existence of OC-free $Fe_d$ (Wang et al., 2019; Faust et
al., 2021). At neutral to alkaline pH, associated with arid and semiarid soils, the
association of reactive Fe and OC is limited (Sowers et al., 2018a; Sowers et al., 2018b),
while calcium (Ca) is especially important in OC binding via Ca bridging (Sowers et
al., 2018a; Wang et al., 2021). Wang et al. (2021) provided direct evidence that the Fe-
OC determined by the classic BCD method contained Ca-bound OC, accounting for
approximately 24% of Fe-OC (Wang et al., 2021), and the Fe-OC:$Fe_d$ molar ratio might,
therefore, be overestimated, for example, in the permafrost regions of the Tibetan
Plateau (soil pH 8.01–9.52) (Mu et al., 2016). Therefore, to draw a valid inference on
the bonding mechanisms of OC and reactive Fe, further work is necessary to unravel
the complex mechanisms.



## 5. Conclusions

To our knowledge, this is the first study to reveal the patterns and drivers of Fe-OC across global ecosystems (Fig. 7). More importantly, our global-scale results showed that Fe-OC was an important fraction of SOC at the continental scale, in wetlands, and in marine ecosystems. Our findings highlighted that some drivers for Fe-OC associations are valid globally, but those ecosystem-specific predictors should also be uncovered. Correlation analysis and RF modelling indicated that the Fe-OC:$Fe_d$ molar ratio and SOC were the predominant predictors of Fe-OC and $f$Fe-OC compared with climate variables and soil pH in global ecosystems. The Fe-OC:$Fe_d$ molar ratio was the predominant driver of Fe-OC at the continental scale and in wetlands, whereas $Fe_d$ content was a good predictor in the global marine ecosystem, improving our ability to predict Fe-OC variations among ecosystem types. Moreover, in global wetlands, the fractions of Fe-OC in total SOC may be increasing in response to climate warming. As an indicator of the Fe and OC bonding mechanism, the molar ratio between 1–6 (<1 for adsorption, >6 for coprecipitation) in global ecosystems exceeds 60%, highlighting the importance of the interactions of both adsorption and coprecipitation. Compared with continental and marine ecosystems, coprecipitation plays a more important role in wetlands due to the high molar ratio. Our findings provide direct evidence that reactive Fe minerals are a dominant natural mechanism for long-term SOC storage in global ecosystems.





**Acknowledgments**

We are very grateful to all the researchers whose data were compiled in this study. This study was supported by the National Natural Science Foundation of China (32101333).

**Conflict of interest**

The authors declare that they have no known competing financial interests or personal relationships that could have appeared to influence the work reported in this paper.

**Data availability statement**

The data that supports the findings of this study are available in the Supporting Data Set.





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

**Figure captions**

**Fig. 1** Global distribution of study sites.
**Fig. 2** The Fe-OC content (a), $f$Fe-OC (b), soil pH (c), $Fe_d$ content (d), SOC content (e),
and Fe-OC/$Fe_d$ molar ratio (f) in different ecosystems shown in the box-plot. Solid dots
indicate outliers, and imaginary points represent observations. Box edges are upper and
lower quartiles; central lines are median value; whiskers represent standard error. The
differences among continental, wetland and marine ecosystems are illustrated (* $p <$
0.05, ** $p < 0.01$, *** $p < 0.001$).
**Fig. 3** Relationships between Fe-OC, $f$Fe-OC and soil properties (soil pH, $Fe_d$, Fe-
OC/$Fe_d$ molar ratio, SOC, clay), climate variables (MAT, MAP) and latitude across
global ecosystem types. The line represents the line of best fit for each ecosystem, and
the shaded area indicates the 95% confidence interval for the global dataset. In marine
ecosystems, the climate variables (MAT, MAP) and soil pH are not shown due to limited
data.
**Fig. 4** The relative importance of climate variables (MAT, MAP), soil properties (SOC,
soil pH, Fe-OC:$Fe_d$, and $Fe_d$), and geographical location (i.e., latitude) for Fe-OC and
$f$Fe-OC in continents (a, b), marine ecosystems (c, d), and wetlands (e, f) by random
forest (RF) analysis. The mean square error (MSE) is used to estimate the importance





of these predictors, with higher MSE values indicating more important predictors. In
marine ecosystems, the climate variables (MAT, MAP) and soil pH are not shown due
to limited data. Ratio: Fe-OC/Fe$_d$ molar ratio. Asterisks show significant differences:
*$p < 0.05$, and **$p < 0.01$.
**Fig. 5** Frequency distributions of the Fe-OC/Fe$_d$ molar ratio in different ecosystems.
The molar ratio of Fe-OC:Fed is used as an indicator of Fe/OC interaction types, which
is $< 1.0$ for adsorption and $> 6$ for coprecipitation (Wagai and Mayer, 2007; Wang et al.,

684    2017).

**Fig. 6** The Spearman correlation analysis results of the Fe-OC, Fe-OC/Fe$_d$ molar ratio
(i.e., ratio) and environmental factors (MAT, MAP, pH) in continental (a), marine (b)
and wetland ecosystems (c). Asterisks show significant differences: *$p < 0.05$, **$p <$
$0.01$, and *** $p < 0.001$.
**Fig. 7** Schematic representation of drivers, dynamic and patterns of Fe-OC associations
in different ecosystem types on global scale. Data are averages of different ecosystem
types. A lower SOC molecular diversity and concomitant lower contents of Fe-OC (e.g.,
sea and continent ecosystems), whereas higher diversity increases the Fe-OC contents
(e.g., wetlands). However, there was no significant difference in the proportion of Fe-
OC in total SOC. The asterisk (*) indicates significant differences.



**Figure 1**

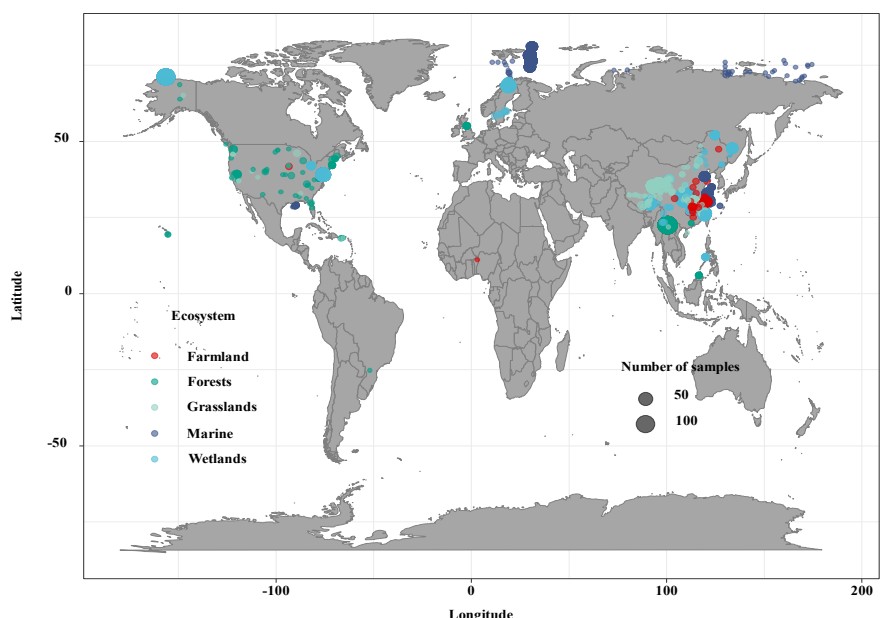






**Figure 2**

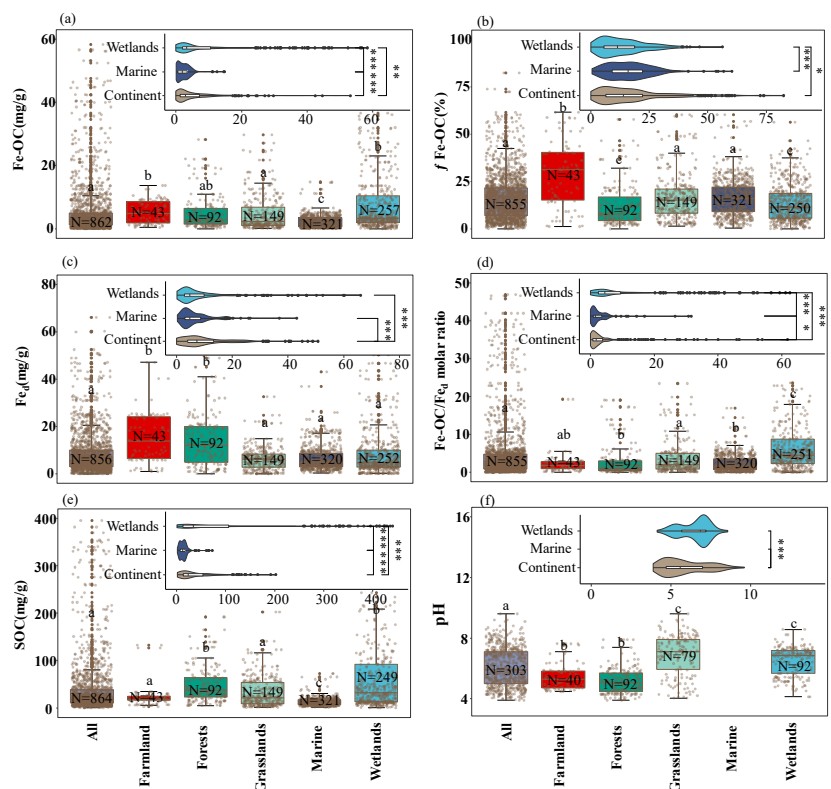





**Figure 3**

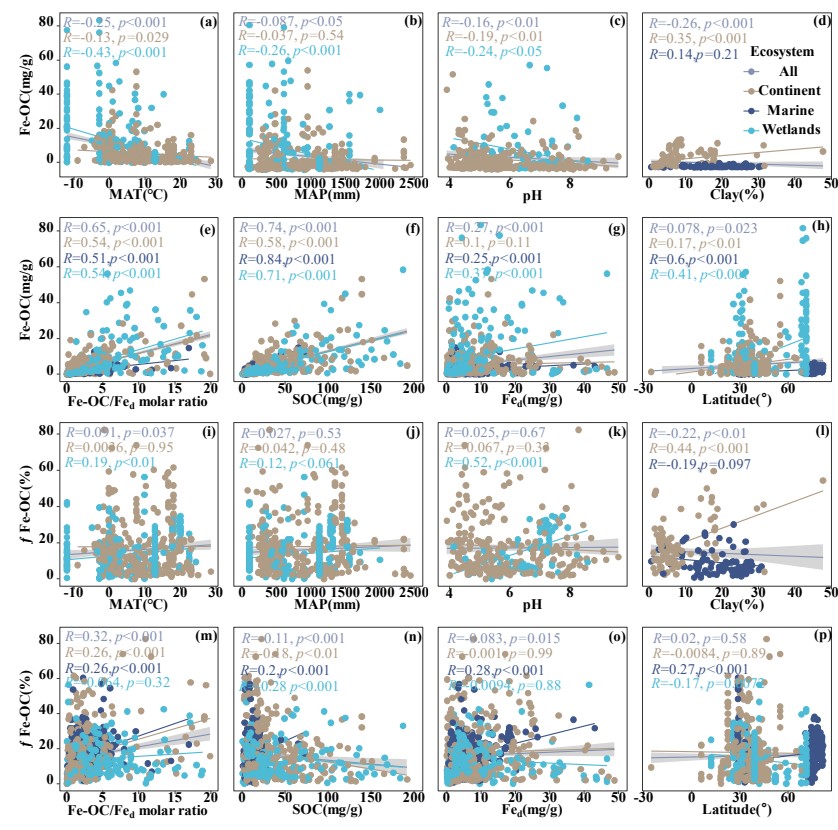






**Figure 4**

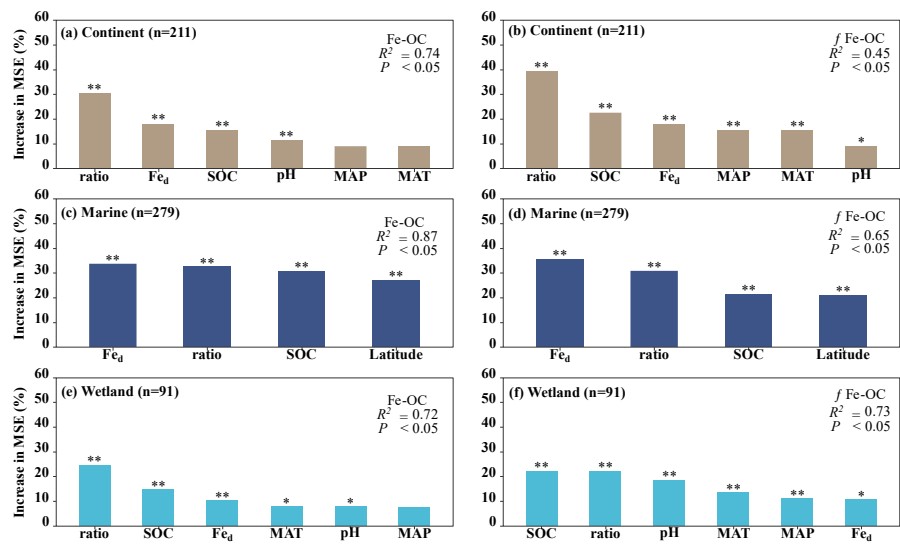






**Figure 5**

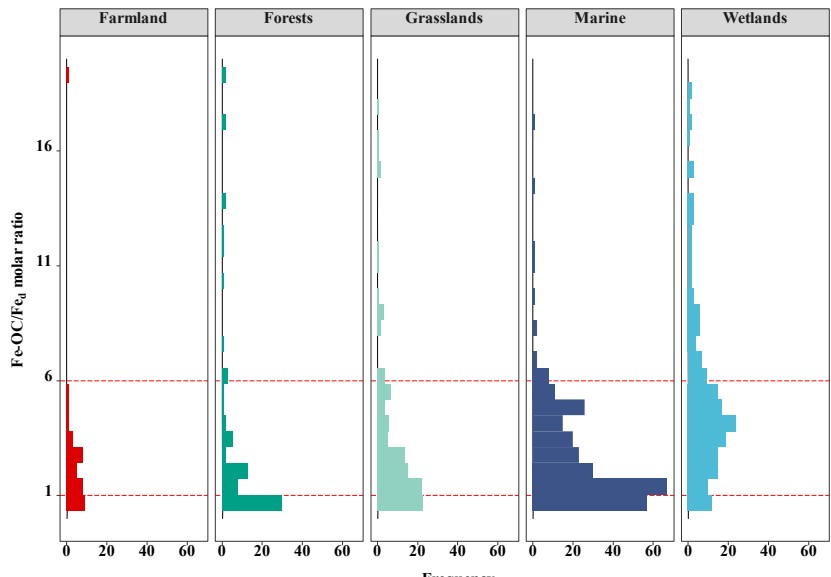






**Figure 6**

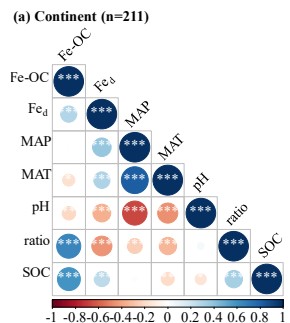 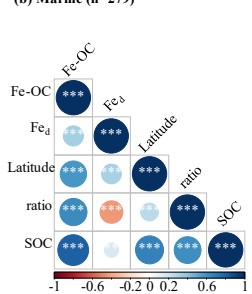 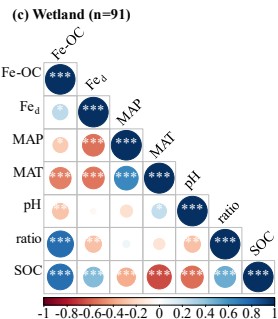






**Figure 7**

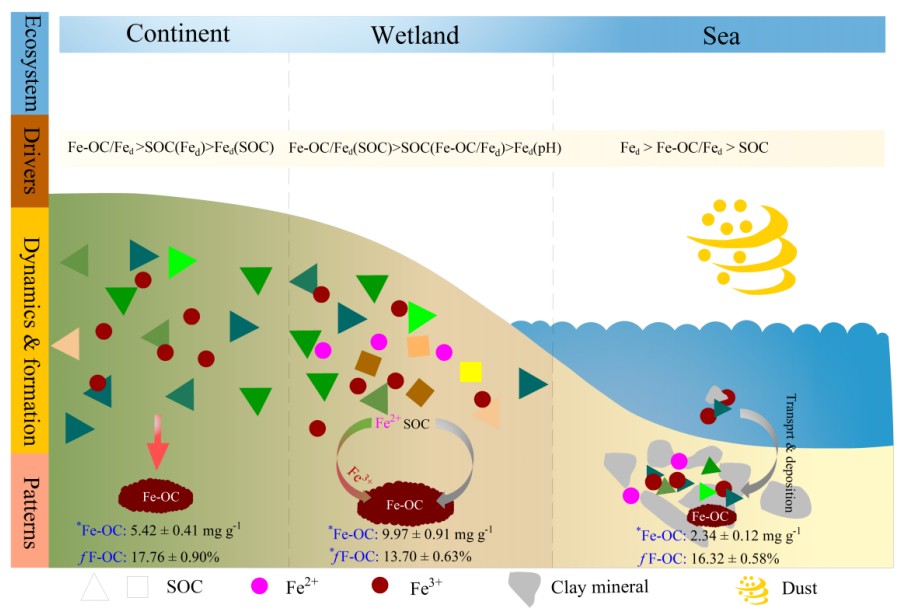
