# Peer review of "Ecosystem-specific patterns and drivers of global reactive iron mineral-associated organic carbon"

_Biogeosciences, 2023_

## Referee Comment (RC1)

The manuscript entitled "Ecosystem-specific patterns and drivers of global reactive iron mineral-associated organic carbon" discussed the spatial variability and characteristics of Fe-OC among continental, wetland and marine ecosystems and its governing factors globally. In this study, they highlighted the importance of reactive Fe oxides in global SOC preservation, and their controlling factors were ecosystem-specific. It is an interesting topic that should fit the scope of the journal. However, minor revision would be needed before a possible consideration of acceptance.

Major comments:

1. The author suggested that Fe-OC/Fed molar ratios less than 1 indicating an Fe-OC bonding form of monolayer surface sorption, and greater than 6 indicating a bonding mechanism dominated by coprecipitation in Line 98-101. So what does the Fe-OC/Fed molar ratio between 1 and 6 mean? Please provide references.

2. Line 153-155: Is the mean annual air temperature extracted from the WordClim database was taken in the same year as the sample collection? Please add something here.

3. Line 164-167: Why did you represent "treatment" and "control" in this way in the manuscript?

4. There are some ambiguous sentences in the manuscript. For example, 1) Since marine ecosystems do not have soil pH, the descriptions may be ambiguous in Line 208-210; 2) "The molar ratio of Fe-OC/Fe$_d$ was positively correlated with Fe-OC and $f$Fe-OC in three ecosystem types, except for $f$Fe-OC in wetlands (Figs. 3e, m)." in Line 224-226. Check other places as well.

5. $f$Fe-OC was significant differences between wetlands and marine ($p < 0.001$) and continent ($p < 0.05$) in Fig. 2b. Why were no significant difference observed in Line 197? However, the authors concluded that significant difference in $f$Fe-OC was observed among different ecosystem types in discussion in Line 281-282. Please specify.

6. How did you estimate the data for SOC storage was bound to Fe oxides in different ecosystems in Line 288-292? Could you provide some calculation process to support this point of view?

7. Line 345-348: To further clarify the complexation of Fe with OC under reducing conditions. Please add the following references:

   Patzner M S, Kainz N, Lundin E, et al. Seasonal fluctuations in iron cycling in thawing permafrost peatlands[J]. Environmental science & technology, 2022, 56(7): 4620-4631.

8. Line 338-339: "……, with <1 suggesting that the OC-Fe bonding form is dominated by simple mono-layer adsorption, while higher ratios indicating coprecipitation." This sentence is confusing. Was the higher ratio >1 or > 6 in this sentence? However, you showed that the molar ratio >6, indicating a bonding mechanism dominated by coprecipitation in the intrdouction. Please specify.

9. Line418-428: It would have been more intuitive to have a table with comparisons to previous studies.

10. Figure7: 1) What do the different coloured triangles and squares represent? 2) There are some symbols and labelling errors. For example, "Sea" should be "marine"; "*f*F-OC" should be "*f*Fe-OC"; 3) Wetlands are not a transitional phase between continent and marine, they are more like a terrestrial ecosystem, so why show them in the middle of continent and marine? Please check other places as well.

Minor comments:

1. Line 15: Fe-associated organic carbon should be abbreviated as OC-Fe not Fe-OC. It should be changed throughout the paper.

2. Please standardize the unit format. For example, "g/kg" (Line 21) should be changed to "g kg$^{-1}$", and the formatting of the units in the article (e.g., mg g$^{-1}$) and in the pictures (e.g., mg/g) was inconsistent.

3. Please check the format of "Fe-OC/dithionite-extractable Fe (Fe$_d$)" (e.g., Line 23) and "Fe-OC:Fe$_d$" (e.g., Line 26).

4. Line 26: Abbreviation for Random Forest.

5. The *f*Fe-OC was repeated in Lines 60 and 75. In addition, the explanation of *f*Fe-OC should be advanced in Line 75.

6. Line 86. Mean annual temperature modified to mean annual air temperature. MAT modified to MAAT. It should be changed throughout the paper.

7. Line 151, Table S1 is not in your manuscript.

8. Line 155: "version" should be "v". In addition, the web address should be added. Check other places as well.

9. Line 222: "Fig. 3O" should be "Fig. 3o".

10. There are many formatting errors, such as the space between a number and a symbol on page 15, Line 287. Please check other places as well.

11. Line 338: Here, it's OC-Fe. Please standardise usage in the manuscript.

12. The letter case of figures in the manuscript should be resized.

13. Figure 2: 1) What does the legends (e.g., a, b, and c) mean? 2) Panels (a), (b), (c), (d), (e), and (f) were shown on different pages. Check other places as well.

14. Figure 3 needs to be re-optimized. For example, *R*, *p* and the image overlap. Check other places as well.

---

## Author Comment (AC1)

**Response of Authors on Manuscript Entitled:** "Ecosystem-specific patterns and drivers of global reactive iron mineral-associated organic carbon (bg-2023-59)"

Dear Editors and Reviewers,

Thank you for your letter and for the reviewers' comments concerning our manuscript entitled "**Ecosystem-specific patterns and drivers of global reactive iron mineral-associated organic carbon** (bg-2023-59)". Those comments are all valuable and very helpful for revising and improving our paper and of important guiding significance to our research. We have studied comments carefully and have made corrections which we hope meet with approval. Revised portion is marked in yellow in the paper. The main corrections in the paper and the responds to the reviewers' comments are as following:

**Response to the First Referee (Reviewer #1):**

(With brown characters as the original comments and suggestions)
* * *
1. The manuscript entitled "Ecosystem-specific patterns and drivers of global reactive iron mineral-associated organic carbon" discussed the spatial variability and characteristics of Fe-OC among continental, wetland and marine ecosystems and its governing factors globally. In this study, they highlighted the importance of reactive Fe oxides in global SOC preservation, and their controlling factors were ecosystem-specific. It is an interesting topic that should fit the scope of the journal. However, minor revision would be needed before a possible consideration of acceptance.

**{Response}** Thank you to the reviewers for their positive comments and feedback. We have individually responded to each comment.

2. The author suggested that Fe-OC/Fed molar ratios less than 1 indicating an Fe-OC bonding form of monolayer surface sorption, and greater than 6 indicating a bonding mechanism dominated by coprecipitation in Line 98-101. So what does the Fe-OC/Fed molar ratio between 1 and 6 mean? Please provide references.

**{Response}** Thank you for your review. We have provided the corresponding reference. Wagai et al. (2007) first pointed out that the maximum adsorption capacity of FeOx phase on OC is 0.22 g-OC g-Fe$^{-1}$ (approximately equal to 1 Fe-OC/Fe$_d$ molar ratios). The mass ratio of OC:Fe ranges from 1.3 to 2.2 (approximately 6-10 Fe-OC/Fed molar ratios), forming organic metal complexes (precipitated complexes). Similarly, Lalonde et al. (2012) further supported and confirmed that the simple sorption of organic matter on reactive iron oxide surfaces results in a maximum molar ratio of organic carbon to iron (OC:Fe) of 1.0 for the co-extracted species, based on the maximal sorption capacity of reactive iron oxides for natural organic matter. However, co-precipitation and/or chelation of organic compounds with iron generates low-density, organic-rich structures with OC:Fe ratios between 6 and 10. Recent study by Wang et al. (2017) suggested that the molar ratio of Fe-bound SOC:Fe$_d$ was used

as an indicator of Fe–OC interaction type, which is <1.0 for sorption and >6.0 for co-precipitation. It is generally accepted that adsorption and coprecipitation coexist when the Fe-OC/$Fe_d$ molar ratio between 1 and 6.

References:

Lalonde, K., Mucci, A., Ouellet, A., Gelinas, Y., 2012. Preservation of organic matter in sediments promoted by iron. Nature 483, 198-200.

Wagai, R., Mayer, L.M., 2007. Sorptive stabilization of organic matter in soils by hydrous iron oxides. Geochimica et Cosmochimica Acta 71, 25-35.

Wang, Y., Wang, H., He, J.S., Feng, X.J., 2017. Iron-mediated soil carbon response to water-table decline in an alpine wetland. Nature Communications 8, 1-9.

3. Line 153-155: Is the mean annual air temperature extracted from the WordClim database was taken in the same year as the sample collection? Please add something here.

{Response} Thank you for your valuable and thoughtful comments. The data obtained from the WorldClim database is the average climate data from 1970 to 2000, which has been supplemented in the article. (see Page 9 Lines 155-156 in the revised manuscript). Furthermore, most articles have reported annual average temperature and precipitation data, with only a small number of literature sources collecting from the WorldClim database.

4. Line 164-167: Why did you represent "treatment" and "control" in this way in the manuscript?

{Response} Thank you for your valuable comments. We will compare the average values of Fe-OC in all articles as a "control", and the average values of individual cases as a "treatments". By calculating the effect size, we can know the difference and contribution between a single case and the overall mean. (see Fig. S2 in the Supporting Materials)

5. There are some ambiguous sentences in the manuscript. For example, 1) Since marine ecosystems do not have soil pH, the descriptions may be ambiguous in Line

208-210; 2) "The molar ratio of Fe-OC/Fed was positively correlated with Fe-OC and $f$Fe-OC in three ecosystem types, except for $f$Fe-OC in wetlands (Figs. 3e, m)." in Line 224-226. Check other places as well.

**{Response}** Thank you for your valuable comments. We have reworded these sentences according to your suggestion. We have changed sentence "Taken together, the Fe-OC, SOC, Fe-OC/Fe$_d$ molar ratio, and soil pH were significantly higher in wetlands, with the lowest values in marine ecosystems across global ecosystem types." to "Taken together, the Fe-OC, SOC, and Fe-OC/Fe$_d$ molar ratio were significantly higher in wetlands, with the lowest values in marine ecosystems across global ecosystem types.". "The molar ratio of Fe-OC/Fe$_d$ was positively correlated with Fe-OC and $f$Fe-OC in three ecosystem types, except for $f$Fe-OC in wetlands" to "The molar ratio of Fe-OC/Fe$_d$ was positively correlated with Fe-OC and $f$Fe-OC in other ecosystems, except for $f$Fe-OC which not correlated with the molar ratios in wetlands.". (see Page 12 Lines 213-214; 229-231 in the revised manuscript)

6. $f$Fe-OC was significant differences between wetlands and marine ($p < 0.001$) and continent ($p < 0.05$) in Fig. 2b. Why were no significant difference observed in Line 197? However, the authors concluded that significant difference in $f$Fe-OC was observed among different ecosystem types in discussion in Line 281-282. Please specify

**{Response}** Thank you for your valuable and thoughtful comments. We have modified it according to your suggestion as follows: "Correspondingly, the contribution of Fe-OC to SOC ($f$Fe-OC) was significantly different among different ecosystem types ($p < 0.05$; Fig. 2b)." (see Page 11 Lines 200-201 in the revised manuscript)

7. How did you estimate the data for SOC storage was bound to Fe oxides in different ecosystems in Line 288-292? Could you provide some calculation process to support this point of view?

**{Response}** Thank you for your valuable comments. We estimated Fe-OC stocks by

multiplying the SOC stock of each ecosystem by the $f$Fe-OC in the corresponding ecosystem. We have added the relevant calculation methods in the Materials and Methods section: "We also collected global data on SOC stocks in terrestrial, wetland and marine ecosystems, respectively, which will allow us to further estimate Fe-OC stocks in different ecosystems". (see Pages 8-9 Lines 145-147 in the revised manuscript)

8. Line 345-348: To further clarify the complexation of Fe with OC under reducing conditions. Please add the following references:

Patzner M S, Kainz N, Lundin E, et al. Seasonal fluctuations in iron cycling in thawing permafrost peatlands[J]. Environmental science & technology, 2022, 56(7): 4620-4631.

**{Response}** We agree with your comments and have cited this reference. We added references: Patzner M S, Kainz N, Lundin E, et al. Seasonal fluctuations in iron cycling in thawing permafrost peatlands. Environmental Science & Technology, 2022, 56(7): 4620-4631. (see Page 18 Line 351 in the revised manuscript)

9. Line 338-339: "……, with <1 suggesting that the OC-Fe bonding form is dominated by simple mono-layer adsorption, while higher ratios indicating coprecipitation." This sentence is confusing. Was the higher ratio >1 or > 6 in this sentence? However, you showed that the molar ratio >6, indicating a bonding mechanism dominated by coprecipitation in the introduction. Please specify.

**{Response}** Thank you for your valuable comments. We have improved the expression according to your suggestion. "Numerous studies have shown that the Fe-OC:Fed acts as an indicator of Fe/OC interaction types (Lalonde et al., 2012; Wang et al., 2017), with <1 suggesting that the Fe-OC bonding form is dominated by simple mono-layer adsorption, while higher molar ratios (>6) indicating coprecipitation (Wagai and Mayer, 2007; Faust et al., 2021)." (see Pages 17-18 Lines 340-344 in the revised manuscript)

References:

Huang, X., Liu, X., Liu, J., Chen, H., 2021. Iron-bound organic carbon and their determinants in peatlands of China. Geoderma 391.

Lalonde, K., Mucci, A., Ouellet, A., Gelinas, Y., 2012. Preservation of organic matter in sediments promoted by iron. Nature 483, 198-200.

Wang, Y., Wang, H., He, J.S., Feng, X., 2017. Iron-mediated soil carbon response to water-table decline in an alpine wetland. Nature Communications 8, 15972.

**10. Line 418-428: It would have been more intuitive to have a table with comparisons to previous studies.**

**{Response}** Thank you for your valuable and thoughtful comments. In accordance with your suggestion, we have added a table about molar ratios of Fe-OC:$Fe_d$ for comparison with previous studies. (see Supporting Data Set in the revised manuscript)

**Table 1.** The molar ratios of Fe-OC:$Fe_d$ across marine sediments and wetland ecosystems.

| Location | Fe-OC:$Fe_d$ | References |
|---|---|---|
| East China Sea | $1.53 \pm 1.28$ | Ma et al., 2018 |
| South Yellow Sea, China | $1.68 \pm 1.80$ | Ma et al., 2018 |
| Bohai Sea, China | $1.59 \pm 1.37$ | Wang et al., 2018 |
| Barents Sea | $2.56 \pm 1.76$ | Faust et al., 2020, 2021 |
| Arctic shelf | $3.04 \pm 1.73$ | Salvadó et al., 2015 |
| Intermediate/old river delta | $5.02 \pm 5.85$ | Shields et al., 2016 |
| Changjiang Estuary/East China Sea shelf | $0.23 \pm 0.14$ | Zhao et al., 2018 |
| Margin sea, China | $2.75 \pm 3.07$ | Sun et al., 2017 |
| Mississippi River | $2.76 \pm 1.50$ | Ghaisas et al., 2021 |
| Global oceans (n = 42) | $6.10 \pm 7.5$ | Lalonde et al., 2012 |
| **Global oceans (n = 320)** | **$2.56 \pm 0.19$** | **This study** |
| Martinique Beach, Canada | $0.4 \pm 0.7$ | Sirois et al., 2018 |
| Min River Estuary, China | $11.0 \pm 4.5$ | Bai et al., 2021 |
| Petland, China | $6.23 \pm 3.34$ | Huang et al., 2021 |
| Freshwater wetland of Sanjiang Plain, China | $2.24 \pm 1.52$ | Duan et al., 2020 |
| Boreal lake sediment | $5.92 \pm 3.34$ | Peter et al., 2018 |
| Permafrost peatland | $0.26 \pm 0.09$ | Wang D et al., 2021 |
| Large-Scale wetlands (19.96˚–52.04˚N, 87.44˚E–132.33˚E) | $12.62 \pm 11.46$ | Wang S et al., 2021 |
| Drained thaw lake basins near Utqiaġvik | $7.40 \pm 5.18$ | Joss et al., 2022 |
| Delmarva Peninsula in the eastern U.S.A | $20.69 \pm 27.89$ | Kottkamp et al., 2022 |

| | | |
|---|---|---|
| Mt. Shen Nong Jia, China | $84.75 \pm 111.95$ | Zhao et al., 2019 |
| **Global wetlands (n = 251)** | **$13.47 \pm 1.81$** | **This study** |

11. Figure7: 1) What do the different coloured triangles and squares represent? 2) There are some symbols and labelling errors. For example, "Sea" should be "marine"; "$f$F-OC" should be "$f$Fe-OC"; 3) Wetlands are not a transitional phase between continent and marine, they are more like a terrestrial ecosystem, so why show them in the middle of continent and marine? Please check other places as well.

**{Response}** Thank you for your valuable and thoughtful comments. We have corrected and improved the Figure7. Different colored triangles and squares represent SOC molecular diversity. We have corrected the symbol and labeling errors as you suggested. Wetlands encompass a broad range of ecosystems ranging from submerged coastal grass beds to salt marshes, swamp forests, and boggy meadows. In this study, the term "wetlands" refers to transition zones between terrestrial and aquatic systems with soils water saturated for at least part of the year or covered by shallow water. In this study, wetland ecosystem included coastal wetlands (for instance, mangrove wetland and tidal wetland) and inland wetlands (for instance, alpine wetland and peatland); Aquatic ecosystem mainly refers to marine and freshwater ecosystems, and the data of freshwater systems in this manuscript are scarce and dominated by marine systems.

[Figure]

**Fig. 7** Schematic representation of drivers, dynamic and patterns of Fe-OC associations in different ecosystem types on global scale. The wetland ecosystem included coastal wetlands and inland wetlands; aquatic ecosystem mainly refers to marine and freshwater ecosystems, but the data of freshwater systems in this study are scarce and dominated by marine systems. Data are averages of different ecosystem types. Different coloured triangles and squares represent SOC molecular diversity. A lower SOC molecular diversity and concomitant lower contents of Fe-OC (e.g., terrestrial and aquatic ecosystems), whereas higher diversity increases the Fe-OC contents (e.g., wetlands). Meanwhile, there was a significant difference in the proportion of Fe-OC in total SOC ($f$Fe-OC). The asterisk (*) indicates significant differences.

---

## Author Comment (AC2)

**Response of Authors on Manuscript Entitled:** "Ecosystem-specific patterns and drivers of global reactive iron mineral-associated organic carbon (bg-2023-59)"

Dear Editors and Reviewers,

Thank you for your letter and for the reviewers' comments concerning our manuscript entitled "**Ecosystem-specific patterns and drivers of global reactive iron mineral-associated organic carbon** (bg-2023-59)". Those comments are all valuable and very helpful for revising and improving our paper and of important guiding significance to our research. We have studied comments carefully and have made corrections which we hope meet with approval. Revised portion is marked in yellow in the paper. The main corrections in the paper and the responds to the reviewers' comments are as following:

**Response to the Second Referee (Reviewer #2)**

(With brown characters as the original comments and suggestions)
* * *
1. The Fe-associated organic carbon is a key component to long-term soil organic carbon (SOC) stocks, making it essential to gain a global understanding of its patterns and drivers. However, there are certain limitations in the current analysis approach. The classification of global ecosystems into continental, wetland, and marine ecosystems lacks precise definitions. Here, "continental" is commonly defined on a geographic scale (e.g., Guerin et al., 2020). Does the term "continental ecosystem" refer to the terrestrial ecosystem?

**{Response}** Thank you for your valuable and thoughtful comments. According to your suggestion, we have revised the "continental ecosystem" to "terrestrial ecosystem" in the whole manuscript. We agree that "continental" is usually defined in terms of geographical scale, so the expression "continental scale" in the article remains.

2. Additionally, the term "wetland ecosystem" in this article includes both freshwater and coastal or estuarine wetland ecosystems. It is important to note that coastal or estuarine wetlands have unique characteristics of salinity. While it is generally believed that Fe complexes may be less affected by increasing salinity, it is dependent on particle size actually (Simon, 2018). Also, salinity can affect Fe transport capacity (Herzog et al., 2019). As such, it is recommended to separate the analysis into freshwater and saline wetlands for more accurate characterization.

**{Response}** Thank you for your valuable and thoughtful comments. We agree that salinity can affect the Fe-OC and $f$Fe-OC. The higher the salt concentration in coastal wetlands, the higher the Fe-OC and $f$Fe-OC (Bai et al., 2021). However, there are few research cases on Fe-OC and $f$Fe-OC in coastal wetlands. Among the 17 wetland articles collected in this article, only four involve coastal wetland (the rest are freshwater wetlands), and the sample size is small. With the increasing number of research articles on Fe-OC in coastal wetlands, it is possible to separate coastal

wetlands from freshwater wetlands in the future to further discuss the effects of salinity on Fe-OC.

References:

Bai, J., Luo, M., Yang, Y., Xiao, S., Zhai, Z., Huang, J., 2021. Iron-bound carbon increases along a freshwater-oligohaline gradient in a subtropical tidal wetland. Soil Biology and Biochemistry 154.

---

## Author Response (AR2)

**Response of Authors on Manuscript Entitled:** "Ecosystem-specific patterns and drivers of global reactive iron mineral-associated organic carbon (bg-2023-59)"

Dear Editor,

Thank you for your letter and for the reviewers' comments concerning our manuscript entitled "**Ecosystem-specific patterns and drivers of global reactive iron mineral-associated organic carbon** (bg-2023-59)". We have studied comments carefully and have made corrections which we hope meet with approval.

**Response to the editor decision:**

(With brown characters as the original comments and suggestions)
* * *
1. line 20, not sure it can be 0 (from 0 to 83.3).

**{Response}** Thank you for your valuable comments. After a comprehensive search of the published literature, we found that there were several articles in which no Fe-OC was detected, or even negative values.

References:

Peter, S., Sobek, S., 2018. High variability in iron-bound organic carbon among five boreal lake sediments. Biogeochemistry 139, 19-29. https://doi.org/10.1007/s10533 -018-0456-8

Wagai, R., Mayer, L.M., 2007. Sorptive stabilization of organic matter in soils by hydrous iron oxides. Geochimica et Cosmochimica Acta 71, 25-35. https://doi.org/10.1016/j.gca.2006.08.047